# An umbrella review of systematic reviews on the impact of the COVID-19 pandemic on cancer prevention and management, and patient needs

Taulant Muka[1,2,3], Joshua JX Li[4], Sahar J Farahani[5], John PA Ioannidis[2,6,7]*

[1]Institute of Social and Preventive Medicine, University of Bern, Bern, Switzerland; [2]Meta-Research Innovation Center at Stanford (METRICS), Stanford University, Stanford, United States; [3]Epistudia, Bern, Switzerland; [4]Department of Anatomical and Cellular Pathology, Prince of Wales Hospital, The Chinese University of Hong Kong, Sha Tin, Hong Kong; [5]Department of Pathology and Laboratory Medicine, Stony Brook University, Long Island, New York, United States; [6]Stanford Prevention Research Center, Department of Medicine, Stanford University School of Medicine, Stanford, United States; [7]Department of Epidemiology and Population Health, Stanford University School of Medicine, Stanford, United States

**Abstract** The relocation and reconstruction of health care resources and systems during the coronavirus disease 2019 (COVID-19) pandemic may have affected cancer care. An umbrella review was undertaken to summarize the findings from systematic reviews on impact of the COVID-19 pandemic on cancer treatment modification, delays, and cancellations; delays or cancellations in screening and diagnosis; psychosocial well-being, financial distress, and use of telemedicine as well as on other aspects of cancer care. Bibliographic databases were searched for relevant systematic reviews with or without meta-analysis published before November 29th, 2022. Abstract, full- text screening, and data extraction were performed by two independent reviewers. AMSTAR-2 was used for critical appraisal of included systematic reviews. Fifty-one systematic reviews were included in our analysis. Most reviews were based on observational studies judged to be at medium and high risk of bias. Only two reviews had high or moderate scores based on AMSTAR-2. Findings suggest treatment modifications in cancer care during the pandemic versus the pre-pandemic period were based on low level of evidence. Different degrees of delays and cancellations in cancer treatment, screening, and diagnosis were observed, with low- and- middle- income countries and countries that implemented lockdowns being disproportionally affected. A shift from in-person appointments to telemedicine use was observed, but utility of telemedicine, challenges in implementation and cost-effectiveness in cancer care were little explored. Evidence was consistent in suggesting psychosocial well-being of patients with cancer deteriorated, and cancer patients experienced financial distress, albeit results were in general not compared to pre-pandemic levels. Impact of cancer care disruption during the pandemic on cancer prognosis was little explored. In conclusion, substantial but heterogenous impact of COVID-19 pandemic on cancer care has been observed.

*For correspondence:
jioannid@stanford.edu

**Competing interest:** The authors declare that no competing interests exist.

## Editor's evaluation

This solid work reviews and synthesizes evidence of the impact of the COVID-19 pandemic on a variety of cancer outcomes. The results have potentially important implications for various fields of

cancer research as they review evidence spanning from cancer prevention efforts to changes in diagnoses and cancer treatment modalities.

## Introduction

The coronavirus disease 2019 (COVID-19) pandemic and the mitigation measures that were undertaken posed major challenges to cancer care. The rapid spread of COVID-19 and early data showing patients with cancer were at increased risk of morbidity and mortality after Severe Acute Respiratory Syndrome Coronavirus 2 (SARS-CoV-2) infection, prompted changes in healthcare delivery (*Venkatesulu et al., 2020*). These changes included reduction of medical activities, reallocation of healthcare workers, shifting in-person appointments to remote consultations, and limiting access of patients to care facilities (*Dhada et al., 2021*).

Concerns have been raised that disruption of healthcare services might have had multidimensional impact in cancer care. Indeed, several studies have described delays and cancellation in treatment, screening, and diagnosis (*Teglia et al., 2022a*; *Teglia et al., 2022b*; *Nikolopoulos et al., 2022*). For example, two meta-analyses showed that during the pandemic there was a ~50% reduction in breast and cervical cancer screening, and that there was 18.7% reduction for all cancer treatments, with surgical treatment showing the highest reduction (*Teglia et al., 2022a*; *Teglia et al., 2022b*). In addition, several studies have highlighted deterioration of psychological well-being of patients with cancer, and psychological, ethical, spiritual, and financial needs of patients with cancer were also affected (*Zhang et al., 2022*; *Kirby et al., 2022*). While several systematic reviews have examined the impact of COVID-19 on cancer care, they evaluated different outcomes and periods of the pandemic, and thus the available review findings are rather fragmented (*Teglia et al., 2022a*; *Teglia et al., 2022b*; *Donkor et al., 2021*; *Gascon et al., 2022*; *Hojaij et al., 2020*; *Legge et al., 2022*; *Murphy et al., 2022*; *Gadsden et al., 2022*; *Majeed et al., 2022*). A comprehensive review of impact of COVID-19 on several aspects of cancer would be essential to understand gaps and scale-up evidence-based interventions, including learning lessons for future pandemics. In addition, although systematic reviews are important for public health and policy decision-making during the pandemic, the level of methodological rigor they implemented is unclear.

In the current study, we performed an umbrella review of systematic reviews to summarize the impact of COVID-19 on several aspects of cancer care, including treatment, diagnosis, financial, psychological, and social dimensions. We assessed the amount and geographical breadth of the available evidence and methodological rigor of the primary studies included in each review (as assessed by the reviewers) and of the systematic reviews themselves; and summarized the conclusions from different reviews on COVID-19 impact.

## Results

Our search strategy identified 1172 citations. Based on title and abstract screening, we retrieved full texts of 96 articles for further screening. Of those, 45 articles did not meet our eligibility criteria, thus leaving 51 articles to be included in our final analysis. *Figure 1* summarizes our screening procedure. No additional study was found from screening of references of the included studies.

### Characteristics of the included systematic reviews

Of the 51 included systematic reviews, 14 articles also included a quantitative analysis/meta-analysis with one being individual participant meta-analysis (*Dhada et al., 2021* ; *Teglia et al., 2022a*; *Teglia et al., 2022b*; *Nikolopoulos et al., 2022*; *Zhang et al., 2022*; *Kirby et al., 2022*; *Donkor et al., 2021*; *Gascon et al., 2022*; *Hojaij et al., 2020*; *Legge et al., 2022*; *Murphy et al., 2022*; *Gadsden et al., 2022*; *Majeed et al., 2022*; *Adham et al., 2022*; *Alom et al., 2021*; *Ayubi et al., 2021*; *Garg et al., 2020*; *Jammu et al., 2021*; *Lu et al., 2021*; *Momenimovahed et al., 2021*; *Mostafaei et al., 2022*; *Moujaess et al., 2020*; *Muls et al., 2022*; *Pacheco et al., 2021*; *Rohilla et al., 2021*; *Salehi et al., 2022*; *Sun et al., 2021*; *Zapała et al., 2022*; *Alkatout et al., 2021*; *Di Cosimo et al., 2022*; *Fancellu et al., 2022*; *Ferrara et al., 2022*; *Hesary and Salehiniya, 2022*; *Lignou et al., 2022*; *Mayo et al., 2021*; *Mazidimoradi et al., 2021*; *Mazidimoradi et al., 2022*; *Ng and Hamilton, 2022*; *Pararas et al., 2022*; *Riera et al., 2021*; *Sarich et al., 2022*; *Sasidharanpillai and Ravishankar,*

**eLife digest** The onset of the COVID-19 pandemic disrupted many aspects of human life, not least healthcare. As resources were redistributed towards the crisis, social isolation rules also limited access to medical professionals. In particular, these measures may have affected many aspects of cancer care, such as early detection or treatment.

Many studies have aimed to capture the impact of these changes, but most have been observational, with researchers recording events without trying to impose a controlled design. These investigations also often faced limitations such as small sample sizes, or only focusing on one aspect of cancer care. Systemic reviews, which synthetize and assess existing research on a topic, have helped to bypass these constraints. However, they are themselves not devoid of biases. Overall, a clear, unified picture of the impact of COVID-19 on cancer care is yet to emerge.

In response, Muka et al. carried an umbrella analysis of 51 systematic reviews on this topic. They used a well-known critical appraisal tool to assess the methodological rigor of each of these studies, while also summarising their findings. This work aimed to capture many aspects of the patients' experience, from diagnosis to treatment and the financial, psychological, physical and social impact of the disease.

The results confirmed that the pandemic had a substantial impact on cancer care, including delays in screening, diagnosis and treatment. Throughout this period cancer patients experienced increased rates of depression, post-traumatic stress and fear of their cancer progressing. The long-term consequences of these disruptions remain to be uncovered.

However, Muka et al. also showed that, overall, these conclusions rely on low-quality studies which may have introduced unaccountable biases. In addition, their review highlights that most of the data currently available has been collected in high- and middle-income countries, with evidence lacking from regions of the world with more limited resources.

In the short-term, these results indicate that interventions may be needed to mitigate the negative impact of the pandemic on cancer care; in the long-term, they also demonstrate the importance of rigorous systematic reviews in guiding decision making. By shining a light on the ripple effects of certain decisions about healthcare resources, this work could also help to shape the response to future pandemics.

*2022*; *Tang et al., 2022*; *Thomson et al., 2020*; *Vigliar et al., 2020*; *de Bock et al., 2022*). Other key characteristics of the 51 systematic reviews are shown in *Table 1* (more extensive details appear in *Supplementary file 1a* and *Supplementary file 2*). The median number of bibliographic databases/data sources that were searched was 3; the most searched databases were PubMed ($n = 35$), Medline ($n = 25$), Embase ($n = 22$), Scopus ($n = 19$), Web of Science ($n = 13$), and The Cumulative Index to Nursing and Allied Health Literature – CINAHL database ($n = 10$). One review searched for mobile applications using the iOS App Store and Android Google Play (*Lu et al., 2021*). The median number of studies included in the systematic reviews was 31 (interquartile range, 15; 51). The type of study designs included across reviews varied, but most reviews included data from observational study designs of cross-sectional and retrospective nature. Twenty-one reviews focused/reported exclusively on studies that include pre-pandemic controls. Twenty reviews provided data only on site-specific cancers, while the rest for any cancer site with or without data on site-specific cancers. Nineteen reviews assessed only one aspect of cancer care, while the rest examined two or more of our pre-defined outcomes. The date of last search varied from April 2020 to May 2022, with 16 reviews ending searches during 2020, 25 during 2021, and 5 during 2022; 4 reviews did not provide information on date of last search.

## Geographical distribution

Out of 51 reviews, 46 provided some information on geographical distribution of the included primary studies. Of those, most reviews provided data from different countries, while only two studies (3.9%) focused on data from India (*Rohilla et al., 2021*) and Italy (*Fancellu et al., 2022*) exclusively. Also the majority of the evidence was derived from high- and middle-income countries.

**Identification of studies via databases and registers**

**Identification**

Records identified from**:
  Pubmed (n=892)
  WHO COVID-19 database (n=280)

Records removed *before screening*:
Duplicate records removed (n=2
)

**Screening**

Records screened
(n=1170)

Records excluded**
(n= 1074)

Reports sought for retrieval
(n=96)

Reports not retrieved
(n=2)

Reports assessed for eligibility
(n= 94)

Reports excluded:
  No outcome of interest (n =10)
  No appropriate study design (n =5)
  Not during COVID-19 and/or no cancer specific results (n = 12)
  No English language (n=2)
  Preprints of conference proceedings (n=14)

**Included**

Studies included in review
(n= 51)

**Figure 1.** Flowchart of identification, screening, eligibility, inclusion, and exclusion of retrieved studies*. *In the search, we did not include any language restriction filter. However, during full-text screening we included only studies that were in English. **WHO COVID-19 database does not allow to specify the search by both date and month, and the search for this specific database is up to end-December 2022. Any full text (*n* = 0) that was eligible and published after November 29, 2022, was excluded.

## Risk of bias of primary studies included in the systematic reviews and GRADE assessments

Of the 51 reviews, 32 assessed risk of bias of the included studies (*Table 2* and details in *Supplementary file 1b*). Thirteen different risks of bias checklists were used, and the most common checklists used to assess methodological rigor were Newcastle-Ottawa Scale (NOS) (*n* = 10) and Joanna Briggs Institute tools (*n* = 7). Of the systematic reviews that assess methodological rigor of the individual studies, 8 concluded strong evidence, 19 mixed evidence, 3 weak evidence, and 2 did not provide any

**Table 1.** Characteristics of included systematic reviews.

| Author, year of publication | Meta-analysis | Number of included studies | Countries* | Pre-pandemic controls | Cancer types | Aspects assessed | Last search |
|---|---|---|---|---|---|---|---|
| *Adham et al., 2022* | No | 5 | Globally | No | H&N | MT, O | 15-Jul-20 |
| *Alkatout et al., 2021* | No | 16 | Multiple countries, including US, TW, BE, NL, JP, IT, UK, AS, CA | Yes | ALL | DCS, RD | 28-Dec-20 |
| *Alom et al., 2021* | No | 72 | Multiple countries | No | All | MT, TL, O | 1-Sep-20 |
| *Ayubi et al., 2021* | Yes | 34 | Multiple countries | No | All | PSND, O | 3-Jan-21 |
| *Azab and Azzam, 2021* | No | 51 | Multiple countries | No | Glioma | MT | End of 2020 |
| *Bezerra et al., 2022* | No | 8 | NP | No | ALL | TL | 01-Apr-2021 |
| *Crosby and Sharma, 2020* | No | 45 | NP | No/NS | H&N | MT | 08-Apr-2020 |
| *de Bock et al., 2022* | Yes | 24 | Multiple countries | Yes | ALL, BC | Delayed and/or canceled treatment Other aspects | 21-Mar-2021 |
| *Dhada et al., 2021* | No | 19 | Multiple countries, including IT, US, UK, NL | No | ALL | DCT, DCS, PSND, TL, FBD, SIA | 1-Dec-20 |
| *Di Cosimo et al., 2022* | Yes | 56 | Multiple countries | Yes | ALL | MT, DCT, TL, O | 11-Dec-20 |
| *Donkor et al., 2021* | No | 11 | Multiple countries, including CN, IR, BR, ZA | No | ALL | O | 3-Aug-20 |
| *Fancellu et al., 2022* | No | 7 | IT | Yes | CRC | DCS, RD | 31-Jan-22 |
| *Ferrara et al., 2022* | No | 33 | Multiple countries | Yes | CV | DCT, DCS, RD, RHPV | 8-Feb-22 |
| *Gadsden et al., 2022* | No | 17 | Multiple countries, including IN, SL, BA | Yes | ALL | DCT, O | 15-Dec-21 |
| *Garg et al., 2020* | No | 212 | Multiple countries | No | ALL | MT | 2-May-20 |
| *Gascon et al., 2022* | No | 23 | Multiple countries | No | H&N | MT, O | 1-May-20 |
| *Hesary and Salehiniya, 2022* | No | 22 | Multiple countries, including IT, UK, PG, NL, CN, IN, JP, TU, IR, SN | Yes | GA | MT, DCS, RD, PSND | 31-Dec-21 |
| *Hojaij et al., 2020* | No | 35 | Multiple countries | No | H&N, OTO | MT, TL, O | 31-Dec-20 |
| *Jammu et al., 2021* | No | 19 | Multiple countries | No | ALL | DCT, PSND, FBD | 27-Aug-20 |
| *Kirby et al., 2022* | No | 56 | Multiple countries | No | ALL | PSND, FBD, SIA | 31-Mar-21 |
| *Legge et al., 2022* | No | 18 | Multiple countries | No | ALL | PSND, FBD, SIA | 25-May-22 |
| *Lignou et al., 2022* | No | 32 | Multiple countries | Yes | PC | DCT, RD, TL | 1-Aug-21 |
| *Lu et al., 2021* | No | 41† | NP | No | ALL | TL | 1-May-20 |
| *Majeed et al., 2022* | No | 60 | Multiple countries | Yes, but NS | PC | DCT, RD, TL | 3-Nov-21 |
| *Mayo et al., 2021* | Yes | 13 | Multiple countries, including IT, AU, TW, US, FR, NL | Yes | ALL | DCT, DCS | 10-Feb-21 |
| *Mazidimoradi et al., 2021* | No | 43 | Multiple countries | Yes | CRC | MT, DCT, RD | 1-Jun-21 |
| *Mazidimoradi et al., 2022* | No | 25 | Multiple countries | Yes | CRC | DCS | 1-Jun-21 |

*Table 1 continued on next page*

*Table 1 continued*

| Author, year of publication | Meta-analysis | Number of included studies | Countries* | Pre-pandemic controls | Cancer types | Aspects assessed | Last search |
|---|---|---|---|---|---|---|---|
| *Momenimovahed et al., 2021* | No | 55 | Multiple countries | No | ALL | PSND | 30-Jun-21 |
| *Mostafaei et al., 2022* | No | 22 | Multiple countries | No | ALL | TL | 1-Jun-21 |
| *Moujaess et al., 2020* | No | 88 | Multiple countries | No | ALL | DCT, O | 15-Apr-20 |
| *Muls et al., 2022* | No | 51 | Multiple countries | No | ALL | PSND | 1-Oct-21 |
| *Murphy et al., 2022* | No | 37 | Multiple countries | No | ALL | TL | 31-Mar-21 |
| *Ng and Hamilton, 2022* | Yes | 31 | Multiple countries | Yes | BC | DCS, RD | 1-Oct-20 |
| *Nikolopoulos et al., 2022* | No | 15 | Multiple countries | Yes, but NS | GC | MT, DCT, RD, PSND | 10-Feb-21 |
| *Pacheco et al., 2021* | No | 9 | Multiple countries, including US, IT, CN, SP, UK, IR | No | ALL | DCT, O | NP |
| *Pararas et al., 2022* | Yes | 10 | Multiple countries | Yes | CRC | O | NP |
| *Pascual et al., 2022* | No | 12 | Multiple countries from low- and middle-income countries | Yes, but NS | Surgical Neuro-Oncology | MD, RD, TL, O | 01-Sep-2021 |
| *Piras et al., 2022* | No | 281 | Multiple countries | No | ALL | MT, DCT, SIA, PSND | 31-Dec-2021 |
| *Riera et al., 2021* | No | 62 | Multiple countries | Yes | ALL | DCT | NP |
| *Rohilla et al., 2021* | No | 6 | IN | No | ALL | PSND, O | 3-Feb-21 |
| *Salehi et al., 2022* | No | 16 | Multiple countries | No | ALL | TL | 1-Apr-21 |
| *Sarich et al., 2022* | Yes | 44 | Multiple countries | Yes | NA | RF | 5-Nov-20 |
| *Sasidharanpillai and Ravishankar, 2022* | Yes | 7 | Multiple countries, including SL, IT, CA, SC, BE, US | Yes | CV | DCT, RD | 1-Sep-21 |
| *Sun et al., 2021* | No | 6 | IT, AM, UK | No | BC | MT | 1-Feb-21 |
| *Tang et al., 2022* | Yes | 14 | TU, CN, UK, IT, DN, AS, AU | Yes | CRC | O | 12-Jan-22 |
| *Teglia et al., 2022a* | Yes | 39 | Multiple countries | Yes | BC, CRC, CV | DCT, RD | 12-Dec-21 |
| *Teglia et al., 2022b* | Yes | 47 | Multiple countries | Yes | ALL | DCT | 12-Dec-21 |
| *Thomson et al., 2020* | Yes | 54 | NP | Yes | ALL | O | 1-Jun-21 |
| *Vigliar et al., 2020* | Yes | 41‡ | Multiple countries | Yes | ALL | DCS, RD | 30-Apr-20 |
| *Zapała et al., 2022* | No | 160 | NP | No | ALL | DCT, PSND, TL | NP |
| *Zhang et al., 2022* | Yes | 40 | Multiple countries | No | ALL | PSND | 31-Jan-22 |

*Multiple countries refer to inclusion of studies for final analysis that used data from more than one country. If complete information on location from all primary studies were provided, then specific countries were listed.

†Apps.

‡Respondents.

AM = America. BC; AS = Austria. AU = Australia. BA = Bangladesh. BC = breast cancer. BE = Belgium. BR = Brazil. CA = Canada. China; CRC = colorectal cancer. CV = cervical cancer. DN = Denmark. FR = France. GA = gastric cancer. GC = gynecological cancer. H&N = head and neck cancer. IN = India. IR = Iran. IT = Italy. JP = Japan. NA = not applicable. NL = Netherlands. NP = not provided. OTO = otorhinolaryngology cancer. PC = pediatric cancer. PG = Portugal. SC = Scotland. SL = Slovenia or Sri Lanka. SN = Singapore. SP = Spain. TU = Turkey. TW = Taiwan. UK = United Kingdom. United States; ZA = Zambia.MT = modification of treatment. DCT = delayed and/or canceled treatment. DCS = delayed and canceled screening. RD = reduced diagnosis: RHPV, reduced uptake of HPV vaccination. TL = telemedicine. PSND = psychological needs/distress. FBD = financial burden/distress. SIA = social isolation. O = other aspects.

**Table 2.** Methodological rigor of included reviews.

| Author | Checklist use | Methodological rigor conclusion category | GRADE |
|---|---|---|---|
| *Adham et al., 2022* | CEBM | Not provided | Not provided |
| *Alkatout et al., 2021* | NOS | Strong evidence | Not provided |
| *Alom et al., 2021* | NHLBI, NIH | Not provided | Not provided |
| *Ayubi et al., 2021* | Not applied | Not provided | Not provided |
| *Azab and Azzam, 2021* | Not applied | Not provided | Not provided |
| *Bezerra et al., 2022* | Not applied | Not provided | Not provided |
| *Di Cosimo et al., 2022* | CLARITY | Mixed/Intermediate | Not provided |
| *Crosby and Sharma, 2020* | Not applied | Not provided | Not provided |
| *de Bock et al., 2022* | ROBINS-I | Strong evidence | Not provided |
| *Dhada et al., 2021* | CASP, NHLBI, NIH | Mixed/Intermediate | Not provided |
| *Donkor et al., 2021* | JBI | Weak | Not provided |
| *Fancellu et al., 2022* | Not applied | Not provided | Not provided |
| *Ferrara et al., 2022* | NOS | Strong evidence | Not provided |
| *Gadsden et al., 2022* | JBI, ROBINS-I | Mixed/Intermediate | Not provided |
| *Garg et al., 2020* | Not applied | Not provided | Not provided |
| *Gascon et al., 2022* | Agree II | Mixed/Intermediate | Not provided |
| *Hesary and Salehiniya, 2022* | NOS | Mixed/Intermediate | Not provided |
| *Hojaij et al., 2020* | Not applied | Not provided | Not provided |
| *Jammu et al., 2021* | Not applied | Not provided | Not provided |
| *Kirby et al., 2022* | JBI, CHEC | Mixed/Intermediate | Not provided |
| *Legge et al., 2022* | MMAT | Strong evidence | Not provided |
| *Lignou et al., 2022* | Not applied | Not provided | Not provided |
| *Lu et al., 2021* | MARS | Mixed/Intermediate | Not provided |
| *Majeed et al., 2022* | Not applied | Not provided | Low to moderate certainty |
| *Mayo et al., 2021* | NOS | Mixed/Intermediate | Moderate to high |
| *Mazidimoradi et al., 2021* | NOS | Mixed/Intermediate | Not provided |
| *Mazidimoradi et al., 2022* | NOS | Strong evidence | Not provided |
| *Momenimovahed et al., 2021* | Not applied | Not provided | Not provided |
| *Mostafaei et al., 2022* | JBI | Mixed/Intermediate | Not provided |
| *Moujaess et al., 2020* | Not applied | Not provided | Not provided |
| *Muls et al., 2022* | MMAT | Mixed/Intermediate | Not provided |
| *Murphy et al., 2022* | JBI, CHEC | Mixed/Intermediate | Not provided |
| *Ng and Hamilton, 2022* | NOS | Mixed/Intermediate | Not provided |
| *Nikolopoulos et al., 2022* | NOS | Mixed/Intermediate | Not provided |
| *Pacheco et al., 2021* | JBI, ROBINS-I | Weak | Not provided |
| *Pararas et al., 2022* | NOS | Strong evidence | Not provided |
| *Pascual et al., 2022* | Not applied | Not provided | Not provided |
| *Piras et al., 2022* | Not applied | Not provided | Not provided |
| *Riera et al., 2021* | ROBINS-I | Mixed/Intermediate | Not provided |

*Table 2 continued on next page*

Table 2 continued

| Author | Checklist use | Methodological rigor conclusion category | GRADE |
|---|---|---|---|
| *Rohilla et al., 2021* | Not applied | Not provided | Not provided |
| *Salehi et al., 2022* | Not applied | Not provided | Not provided |
| *Sarich et al., 2022* | ROBINS-I | Weak evidence | Not provided |
| *Sasidharanpillai and Ravishankar, 2022* | NHLBI, NIH | Strong evidence | Not provided |
| *Sun et al., 2021* | Not applied | Not provided | Not provided |
| *Tang et al., 2022* | NOS | Strong evidence | Not provided |
| *Teglia et al., 2022a* | CASP | Mixed/Intermediate | Not provided |
| *Teglia et al., 2022b* | CASP | Mixed/Intermediate | Not provided |
| *Thomson et al., 2020* | ASTRO | Mixed/Intermediate | Not provided |
| *Vigliar et al., 2020* | Not applicable | Not provided | Not provided |
| *Zapała et al., 2022* | Not applied | Not provided | Not provided |
| *Zhang et al., 2022* | JBI | Mixed/Intermediate | Not provided |

CEBM, Critical appraisal tool of qualitative studies from Centre of Evidence-based Medicine (CEBM), University of Oxford; ASTRO, The American Society of Radiation Oncology; CASP, https://casp-uk.net/casp-tools-checklists/; CHEC, Consensus on Health Economic Criteria: CLARITY, 'Risk of bias instrument for cross-sectional surveys of attitudes and practices' from the CLARITY Group at McMaster University; JBI, Joanna Briggs Institute; MARS, Mobile Apps Rating Scale; MMAT, Mixed Methods Appraisal Tool; NHLBI, NHI, National Institute of Health Checklist; NOS, Newcastle-Ottawa Quality Assessment: RBC, Risk of Bias Checklist for Prevalence Studies by **Hoy et al., 2012**.

results. Excluding the NOS assessments [since NOS has been criticized to not provide accurate assessment of methodological rigor (**Stang, 2010**)], the respective numbers were 3, 14, 3, and 2. Only two reviews used GRADE (Grading of Recommendations, Assessment, Development and Evaluations), concluding low to moderate certainty in the results.

## Methodological rigor of included systematic reviews

*Table 3* shows the AMSTAR-2 evaluations for the included systematic reviews. Only two reviews scored moderate to high quality, while the rest were evaluated as low or critically low quality due to not meeting one or more of the seven domains considered critical. Most of the studies did not provide the list of excluded studies during the full-text screening, and did not account for methodological rigor of included studies when interpreting/discussing the results of the reviews.

## Results and conclusions of systematic reviews and of meta-analyses

The main results and conclusions of the eligible systematic reviews are presented in *Supplementary file 1c-j* for various aspects of cancer care. *Table 4* lists the effect sizes and confidence intervals (CIs) for the systematic reviews that used formal meta-analysis as well as heterogeneity metrics. *Figure 2* provides a summary of main findings of this umbrella review. Here, we present some key findings for each type of outcome:

### Modification of treatment

There were 15 reviews assessing modification of treatment (*Nikolopoulos et al., 2022*; *Gascon et al., 2022*; *Hojaij et al., 2020*; *Adham et al., 2022*; *Alom et al., 2021*; *Garg et al., 2020*; *Moujaess et al., 2020*; *Sun et al., 2021*; *Di Cosimo et al., 2022*; *Hesary and Salehiniya, 2022*; *Mazidimoradi et al., 2021*; *Crosby and Sharma, 2020*; *Azab and Azzam, 2021*; *Pascual et al., 2022*; *Piras et al., 2022*). Main findings for each individual review are outlined in *Supplementary file 1c* and *Table 4*. All reviews were consistent reporting changes in treatment, with downscaling treatments plans in patients with cancer being a significant intervention. *Di Cosimo et al., 2022* reported changes in treatment plans in 65% (95% CI, 53–75%; $I^2$, 98%) of centers (*Di Cosimo et al., 2022*). Guidelines recommended use of non-surgical treatment over surgical treatments, as it was seen in head and neck cancer management. However, reviews suggested patients being assessed in a case-by-case basis and that individual

**Table 3.** Methodological assessment of the included reviews – AMSTAR-2 evaluation (16 questions)*.

| Authors, year of publication | Q1 | Q2 | Q3 | Q4 | Q5 | Q6 | Q7 | Q8 | Q9† | Q10 | Q11 | Q12 | Q13 | Q14 | Q15 | Q16 | Overall assessment |
|---|---|---|---|---|---|---|---|---|---|---|---|---|---|---|---|---|---|
| Adham et al., 2022 | n | n | n | py | n | n | n | n | y | n | na | na | na | n | na | n | Critical low |
| Alkatout et al., 2021 | n | py | y | py | n | n | n | py | y | n | na | na | n | n | na | y | Critical low |
| Alom et al., 2021 | n | n | n | py | n | y | n | py | y | n | na | na | y | n | na | y | Critical low |
| Ayubi et al., 2021 | y | n | n | py | n | n | n | y | n | n | y | n | n | n | y | y | Critical low |
| Azab and Azzam, 2021 | n | n | n | py | y | y | n | y | py | n | y | n | n | n | y | y | Critical low |
| Bezerra et al., 2022 | y | n | n | n | n | n | n | y | n | n | na | na | n | n | na | y | Critical low |
| Crosby and Sharma, 2020 | n | n | n | n | n | n | n | n | n | n | na | na | na | n | na | y | Critical low |
| de Bock et al., 2022 | y | n | y | py | y | y | n | y | y | n | y | n | n | y | n | y | Critical low |
| Dhada et al., 2021 | n | py | n | py | n | n | n | y | y | n | na | na | n | n | na | y | Critical low |
| Di Cosimo et al., 2022 | n | n | n | py | y | n | n | y | y | n | y | y | y | y | y | y | Critical low |
| Donkor et al., 2021 | n | n | n | py | y | y | n | y | y | n | na | na | na | n | na | y | Critical low |
| Fancellu et al., 2022 | y | n | n | n | n | n | n | n | n | n | na | na | n | n | n | n | Critical low |
| Ferrara et al., 2022 | n | py | n | py | y | y | n | n | y | n | na | na | y | n | na | y | Low |
| Gadsden et al., 2022 | y | py | n | py | y | n | n | y | y | n | na | na | y | n | na | y | Low |
| Garg et al., 2020 | n | n | n | py | y | y | n | n | n | n | na | na | n | y | na | y | Critical low |
| Gascon et al., 2022 | y | y | n | y | y | y | n | na | y | y | na | na | na | n | na | y | Low |
| Hesary and Salehiniya, 2022 | n | py | n | py | n | n | n | n | y | n | na | na | n | n | na | y | Critical low |
| Hojaij et al., 2020 | n | n | n | n | n | n | n | n | n | n | na | na | na | n | na | y | Critical low |
| Jammu et al., 2021 | n | n | n | py | y | y | n | n | n | n | na | na | n | n | na | y | Critical low |
| Kirby et al., 2022 | y | py | n | y | n | y | n | py | y | n | na | na | n | n | na | y | Critical low |
| Legge et al., 2022 | y | py | y | py | y | y | n | y | y | n | na | na | n | n | na | y | Critical low |
| Lignou et al., 2022 | y | n | n | n | y | n | n | y | n | n | na | na | n | n | na | y | Critical low |
| Lu et al., 2021 | y | n | na | py | n | n | n | y | na | n | na | na | na | n | na | y | Critical low |
| Majeed et al., 2022 | n | y | n | py | n | n | n | n | py | n | na | na | n | n | na | y | Critical low |
| Mayo et al., 2021 | n | y | n | py | y | y | n | n | py | n | n | y | y | n | n | y | Critical low |
| Mazidimoradi et al., 2022 | n | py | n | py | n | n | n | py | y | n | na | na | n | n | na | y | Critical low |
| Mazidimoradi et al., 2021 | n | py | n | py | n | n | n | y | y | n | na | na | n | n | na | y | Critical low |
| Momenimovahed et al., 2021 | n | n | n | py | n | n | n | n | n | n | na | na | n | n | na | y | Critical low |
| Mostafaei et al., 2022 | n | py | n | n | n | n | y | py | y | n | na | na | n | n | na | y | Critical low |
| Muls et al., 2022 | y | py | y | py | n | n | n | y | y | n | na | na | n | n | na | y | Critical low |
| Murphy et al., 2022 | n | n | n | y | n | n | n | y | y | n | na | na | n | n | na | y | Critical low |
| Ng and Hamilton, 2022 | n | py | n | py | n | n | n | py | y | n | y | n | y | y | y | y | Low |
| Nikolopoulos et al., 2022 | n | py | n | py | n | n | n | n | y | n | na | na | n | n | na | y | Critical low |
| Pacheco et al., 2021 | y | y | y | py | y | y | y | py | y | y | na | na | y | n | na | y | High quality |
| Pararas et al., 2022 | n | y | n | y | y | n | n | n | y | n | n | n | n | y | y | y | Critical low |
| Pascual et al., 2022 | y | n | y | py | y | y | n | y | n | n | na | na | n | y | na | n | Critical low |
| Piras et al., 2022 | n | n | n | py | n | n | n | py | y | n | na | na | n | n | na | y | Critical low |
| Riera et al., 2021 | n | py | y | py | y | y | y | y | y | y | na | na | n | y | na | y | Moderate quality |
| Rohilla et al., 2021 | n | n | n | py | n | y | n | n | n | n | na | na | n | n | na | y | Critical low |

*Table 3 continued on next page*

Table 3 continued

| Authors, year of publication | Q1 | Q2 | Q3 | Q4 | Q5 | Q6 | Q7 | Q8 | Q9† | Q10 | Q11 | Q12 | Q13 | Q14 | Q15 | Q16 | Overall assessment |
|---|---|---|---|---|---|---|---|---|---|---|---|---|---|---|---|---|---|
| *Salehi et al., 2022* | n | n | n | py | y | n | n | n | n | n | na | na | n | n | na | y | Critical low |
| *Sarich et al., 2022* | y | y | y | py | y | y | n | y | y | n | y | y | n | y | n | y | Critical low |
| *Sasidharanpillai and Ravishankar, 2022* | n | py | n | py | n | n | n | y | y | n | y | y | y | y | y | y | Low |
| *Sun et al., 2021* | n | n | n | py | n | n | n | n | n | n | na | na | na | n | na | n | Critical low |
| *Tang et al., 2022* | y | n | n | n | n | n | n | n | y | py | n | n | n | y | n | y | Critical low |
| *Teglia et al., 2022a* | y | py | y | py | y | y | n | n | y | n | n | n | n | n | y | y | Critical low |
| *Teglia et al., 2022b* | y | py | y | py | y | y | n | py | y | n | n | n | n | y | n | y | Critical low |
| *Thomson et al., 2020* | n | n | n | n | n | n | n | n | y | n | y | n | n | n | na | y | Critical low |
| *Vigliar et al., 2020†* | na | na | na | na | na | na | na | na | na | na | na | na | na | na | na | na | NA |
| *Zapała et al., 2022* | n | n | n | n | n | n | n | n | n | n | na | na | n | n | na | y | Critical low |
| *Zhang et al., 2022* | y | y | y | py | n | y | n | py | y | n | y | y | y | y | y | y | Low |

AMSTAR-2 overall assessment rating: high—the review provides an accurate and comprehensive summary of the results of the available studies that addresses the question of interest; moderate—the review has more than one weakness, but no critical flaws. It may provide an accurate summary of the results of the available studies; low—the review has a critical flaw and may not provide an accurate and comprehensive summary of the available studies that address the question of interest; or critically low—the review has more than one critical flaw and should not be relied on to provide an accurate and comprehensive summary of the available studies.

Q1: Did the research questions and inclusion criteria for the review include the components of PICO?

Q2: Did the report of the review contain an explicit statement that the review methods were established prior to the conduct of the review and did the report justify any significant deviations from the protocol?

Q3: Did the review authors explain their selection of the study designs for inclusion in the review?

Q4: Did the review authors use a comprehensive literature search strategy?

Q5: Did the review authors perform study selection in duplicate?

Q6: Did the review authors perform data extraction in duplicate?

Q7: Did the review authors provide a list of excluded studies and justify the exclusions?

Q8: Did the review authors describe the included studies in adequate detail?

Q9: Did the review authors use a satisfactory technique for assessing the risk of bias (RoB) in individual studies that were included in the review?

Q10: Did the review authors report on the sources of funding for the studies included in the review?

Q11: If meta-analysis was performed did the review authors use appropriate methods for statistical combination of results?

Q12: If meta-analysis was performed, did the review authors assess the potential impact of RoB in individual studies on the results of the meta-analysis or other evidence synthesis?

Q13: Did the review authors account for RoB in individual studies when interpreting/discussing the results of the review?

Q14: Did the review authors provide a satisfactory explanation for, and discussion of, any heterogeneity observed in the results of the review?

Q15: If they performed quantitative synthesis did the review authors carry out an adequate investigation of publication bias (small study bias) and discuss its likely impact on the results of the review?

Q16: Did the review authors report any potential sources of conflict of interest, including any funding they received for conducting the review?

*The review scored yes if study used a checklist to evaluate methodological rigor, and partial yes if only GRADE assessment was provided without applying a checklist for assessing methodological rigor.

†Individual participant meta-analysis and thus not applicable the AMSTAR evaluation.

n = no. na = not applicable. py = partially yes. y = yes.

factors should be considered for individualized treatment (*Supplementary file 1c*). *Garg et al., 2020* found that available guidelines were based on low level of evidence and had significant discordance for the role and timing of surgery, especially in early tumors (*Garg et al., 2020*).

## Delayed and/or canceled treatment

*Supplementary file 1d* and *Table 4* summarize the main findings from the 15 reviewes (*Dhada et al., 2021*; *Teglia et al., 2022b*; *Nikolopoulos et al., 2022*; *Gadsden et al., 2022*; *Majeed et al., 2022*; *Jammu et al., 2021*; *Pacheco et al., 2021*; *Zapała et al., 2022*; *Di Cosimo et al., 2022*; *Ferrara et al., 2022*; *Lignou et al., 2022*; *Mazidimoradi et al., 2021*; *Riera et al., 2021*; *de Bock et al., 2022*; *Piras et al., 2022*) that assessed and reported on treatment delays and cancellations of

**Table 4.** Summary estimates of the meta-analysis included.

| Author | No. of studies | Outcome | Estimate | LCI | UCI | $I^2$ | p-heterogeniety | Metric |
|---|---|---|---|---|---|---|---|---|
| *Ayubi et al., 2021* | 15 | Depression | 0.37 | 0.27 | 0.47 | 99 | <0.001 | Prev[†] |
| | 17 | Anxiety | 0.38 | 0.31 | 0.46 | 99 | <0.001 | Prev[†] |
| | 4 | Anxiety | 0.25 | 0.08 | 0.42 | 68 | 0.02 | SMD[†] |
| *Zhang et al., 2022* | 28 | Depression | 0.325 | 0.263 | 0.392 | 99 | <0.001 | Prev[†] |
| | 34 | Anxiety | 0.313 | 0.254 | 0.375 | 99 | <0.001 | Prev[†] |
| | 8 | PTSD | 0.288 | 0.207 | 0.368 | 99 | <0.001 | Prev[†] |
| | 5 | Distress | 0.539 | 0.469 | 0.609 | 67 | 0.016 | Prev[†] |
| | 5 | Insomia | 0.232 | 0.171 | 0.293 | 91 | <0.001 | Prev[†] |
| | 3 | Fear of cancer progression | 0.674 | 0.437 | 0.91 | 93 | <0.001 | Prev[†] |
| *Di Cosimo et al., 2022* | 28 | Cancellation/delay of treatment | 0.58 | 0.48 | 0.67 | 98 | <0.01 | Prop*[†] |
| | 14 | Modification of treatment | 0.65 | 0.53 | 0.75 | 98 | <0.01 | Prop*[†] |
| | 10 | Delay of clinic visits | 0.75 | 0.49 | 0.95 | 99 | <0.01 | Prop*[†] |
| | 14 | Reduction in activity | 0.58 | 0.47 | 0.68 | 93 | <0.01 | Prop*[†] |
| | 25 | Use of remote consultation | 0.72 | 0.59 | 0.84 | 99 | <0.01 | Prop*[†] |
| | 7 | Routine use of PPE (patients) | 0.81 | 0.75 | 0.95 | 96 | <0.01 | Prop*[†] |
| | 16 | Routine use of PPE (workers) | 0.8 | 0.61 | 0.94 | 99 | <0.01 | Prop*[†] |
| | 18 | Routine screening SARA-CoV-2 swab | 0.41 | 0.3 | 0.53 | 96 | <0.01 | Prop*[†] |
| | 5 | ≥T2 stage during the COVID-19 pandemic compared to the pre-pandemic control group | 1.00 | 0.72 | 1.38 | 58 | 0.05 | OR[‡] |
| | 4 | ≥T3 stage during the COVID-19 pandemic compared to the pre-pandemic control group | 0.95 | 0.69 | 1.32 | 39 | 0.18 | OR[‡] |
| *de Bock et al., 2022* | 5 | ≥N1 stage during the COVID-19 pandemic compared to the pre-pandemic control group | 1.55 | 0.87 | 2.74 | 3 | 0.39 | OR[‡] |
| *Mayo et al., 2021* | 6 | Screening breast cancer | 0.63 | 0.53 | 0.77 | 100 | <0.001 | IRR[‡] |
| | 5 | Screening conlonc cancer | 0.11 | 0.05 | 0.24 | 100 | <0.001 | IRR[‡] |
| | 3 | Screening cervical cancer | 0.1 | 0.04 | 0.24 | 100 | <0.001 | IRR[‡] |
| *Ng and Hamilton, 2022* | 3 | Screening breast cancer registry-based study | 0.59 | 0.46 | 0.7 | 100 | <0.001 | RR[‡] |
| | 10 | Screening breast cancer non-registry-based study | 0.47 | 0.38 | 0.58 | 100 | <0.001 | RR[‡] |
| | 4 | Diagnosis breast cancer registry-based study | 0.82 | 0.63 | 1.06 | 99 | <0.001 | RR[‡] |
| | 18 | Diagnosis breast cancer non-registry-based study | 0.71 | 0.63 | 0.8 | 92 | <0.001 | RR[‡] |

*Table 4 continued on next page*

*Table 4 continued*

| Author | No. of studies | Outcome | Estimate | LCI | UCI | $I^2$ | p-heterogeniety | Metric |
|---|---|---|---|---|---|---|---|---|
| *Pararas et al., 2022* | 5 | Tis-T1 stage | 1.14 | 0.87 | 1.48 | 41 | 0.15 | OR‡ |
| | 5 | T2 stage | 0.91 | 0.78 | 1.06 | 0 | 0.6 | OR‡ |
| | 5 | T3 stage | 1.18 | 0.82 | 1.7 | 88 | <0.001 | OR‡ |
| | 6 | T4 stage | 1.19 | 0.79 | 1.8 | 80 | <0.001 | OR‡ |
| | 6 | N+ stage | 1 | 0.89 | 1.11 | 0 | 0.54 | OR‡ |
| | 6 | M+ stage | 1.65 | 1.02 | 2.67 | 91 | <0.001 | OR‡ |
| | 7 | Right-sided tumors | 0.88 | 0.51 | 1.52 | 99 | <0.001 | OR‡ |
| | 7 | Left-sided tumors | 0.91 | 0.56 | 1.5 | 96 | <0.001 | OR‡ |
| | 8 | Rectal tumors | 0.93 | 0.63 | 1.37 | 95 | <0.001 | OR‡ |
| | 3 | Emergency presantations | 1.74 | 1.07 | 2.84 | 95 | <0.001 | OR‡ |
| | 3 | Complicated tumor | 1.72 | 0.78 | 3.78 | 82 | 0.004 | OR‡ |
| | 3 | Neoadjuvant therapy | 1.22 | 1.09 | 1.37 | 0 | 0.4 | OR‡ |
| | 4 | Palliative internt surgery | 1.95 | 1.13 | 3.36 | 54 | 0.09 | OR‡ |
| | 6 | Minimally invasive surgery | 0.68 | 0.37 | 1.24 | 98 | <0.001 | OR‡ |
| | 5 | Stoma formation | 0.91 | 0.51 | 1.62 | 94 | <0.001 | OR‡ |
| | 2 | Morbidity | 0.92 | 0.55 | 1.55 | 25 | 0.25 | OR‡ |
| | 3 | Leng of hospital stay | 0.51 | −0.93 | 1.94 | 79 | 0.008 | WMD‡ |
| | 3 | Lymph node harvest | 1.57 | −1.99 | 5.13 | 64 | 0.06 | WMD‡ |
| *Sarich et al., 2022* | 12 | Smoking prevalence | 0.87 | 0.79 | 0.97 | 99 | <0.001 | PR‡ |
| | 17 | Among smokers, smoking less prevalence | 0.21 | 0.14 | 0.3 | 99 | <0.001 | Prev† |
| | 22 | Among smokers, smoking more | 0.27 | 0.22 | 0.32 | 98 | <0.001 | Prev† |
| | 17 | Among smokers, smoking unchanged | 0.5 | 0.41 | 0.58 | 99 | <0.001 | Prev† |
| | 6 | Among smokers, quit smoking | 0.04 | 0.01 | 0.09 | 95 | <0.001 | Prev† |
| | 4 | Among non-smokers, started smoking | 0.02 | 0.01 | 0.03 | 92 | <0.001 | Prev† |
| *Sasidharanpillai and Ravishankar, 2022* | 7 | Women screened before the COVID-19 pandemic | 0.0979 | 0.06 | 0.1359 | 100 | <0.001 | Prop |
| | 7 | Women screened during the COVID-19 pandemic | 0.0424 | 0.0277 | 0.0571 | 100 | <0.001 | Prop |
| *Tang et al., 2022* | 10 | Postoperative morbidity | 0.9 | 0.8 | 1.01 | 26 | 0.22 | OR‡ |
| | 8 | Postoperative mortality | 1.27 | 0.92 | 1.75 | 0 | 0.57 | OR‡ |
| | 4 | Converion rate | 1.07 | 0.75 | 1.52 | 31 | 0.23 | OR‡ |
| | 5 | Incidence of anastomotic leakage | 0.71 | 0.07 | 19.22 | 0 | 0.74 | OR‡ |
| | 2 | Intensive care unit demand rate | 0.73 | 0.29 | 1.85 | 0 | 0.5 | OR‡ |
| | 4 | R1 resections rate | 0.46 | 0.11 | 1.9 | 0 | 0.48 | OR‡ |
| | 5 | Mean lymph node yield | 0.16 | −2.26 | 2.59 | 54 | 0.07 | MD‡ |
| | 7 | Length of hospital stay | −0.05 | −2.28 | 2.19 | 98 | <0.001 | MD‡ |

*Table 4 continued*

| Author | No. of studies | Outcome | Estimate | LCI | UCI | $I^2$ | p-heterogeniety | Metric |
|---|---|---|---|---|---|---|---|---|
| *Teglia et al., 2022a* | 21 | Breast cancer screening January–October 2020 | 0.467 | 0.378 | 0.378 | NP | NP | PRED‡ |
| | 21 | Breast cancer screening April 2020 | 0.74 | 0.567 | 0.918 | NP | NP | PRED‡ |
| | 21 | Breast cancer screening June–October 2020 | 0.13 | −0.07 | 0.33 | NP | NP | PRED‡ |
| | 22 | Colorectal cancer screening January–October 2020 | 0.449 | 0.361 | 0.538 | NP | NP | PRED‡ |
| | 21 | Colonoscopy screening January–October 2020 | 0.525 | 0.388 | 0.663 | NP | NP | PRED‡ |
| | 21 | Fecal occult blood test or fecal immunochemical test January–October 2020 | 0.378 | 0.258 | 0.499 | NP | NP | PRED‡ |
| | 21 | Colorectal cancer screening April 2020 | 0.693 | 0.369 | 1 | NP | NP | PRED‡ |
| | 21 | Colorectal cancer screening June–October 2020 | 0.234 | 0.024 | 0.444 | NP | NP | PRED‡ |
| | 11 | Cervical cancer screening January–October 2020 | 0.518 | 0.389 | 0.647 | NP | NP | PRED‡ |
| | 21 | Cervical cancer screening March 2020 | 0.788 | 0.583 | 0.993 | NP | NP | PRED‡ |
| | | | | | | | | PRED‡ |
| *Teglia et al., 2022b* | NP | Overall treatment January–October 2020 | 0.187 | 0.133 | 0.241 | NP | NP | PRED‡ |
| | NP | Overall treatment January–February 2020 | 0.027 | 0.045 | 0.1 | NP | NP | PRED‡ |
| | NP | Overall treatment March 2020 | 0.156 | 0.076 | 0.237 | NP | NP | PRED‡ |
| | NP | Overall treatment April 2020 | 0.283 | 0.194 | 0.372 | NP | NP | PRED‡ |
| | NP | Overall treatment May 2020 | 0.262 | 0.176 | 0.041 | NP | NP | PRED‡ |
| | NP | Overall treatment June–October 2020 | 0.16 | 0.041 | 0.279 | NP | NP | PRED‡ |
| | NP | Overall surgical treatment January–October 2020 | 0.339 | 0.279 | 0.399 | NP | NP | PRED‡ |
| | NP | Overall surgical treatment January–February 2020 | 0.072 | −0.093 | 0.238 | NP | NP | PRED‡ |
| | NP | Overall surgical treatment March 2020 | 0.307 | 0.219 | 0.396 | NP | NP | PRED‡ |
| | NP | Overall surgical treatment April 2020 | 0.342 | 0.239 | 0.445 | NP | NP | PRED‡ |
| | NP | Overall surgical treatment May 2020 | 0.416 | 0.318 | 0.514 | NP | NP | PRED‡ |
| | NP | Overall surgical treatment June–October 2020 | 0.351 | 0.186 | 0.516 | NP | NP | PRED‡ |
| | NP | Overall medical treatment January–October 2020 | 0.126 | 0.048 | 0.204 | NP | NP | PRED‡ |
| | NP | Overall medical treatment January–February 2020 | 0.015 | −0.055 | 0.084 | NP | NP | PRED‡ |
| | NP | Overall medical treatment March 2020 | 0.116 | −0.012 | 0.233 | NP | NP | PRED‡ |
| | NP | Overall medical treatment April 2020 | 0.248 | 0.09 | 0.407 | NP | NP | PRED‡ |
| | NP | Overall medical treatment May 2020 | 0.196 | 0.085 | 0.306 | NP | NP | PRED‡ |

*Table 4 continued*

| Author | No. of studies | Outcome | Estimate | LCI | UCI | I² | p-heterogeniety | Metric |
|---|---|---|---|---|---|---|---|---|
| | NP | Overall medical treatment June–October 2020 | 0.079 | −0.078 | 0.236 | NP | NP | PRED‡ |
| | | | | | | | | PRED‡ |
| *Vigliar et al., 2020* | 41 | Cytological samples over 4 weeks of the COVID-19 pandemic | 0.453 | 0.001 | 0.98 | NP | NP | PRED‡ |
| | 41 | Ratio of exfoliative to fine needle aspiration samples | 0.89 | 0.74 | 1.08 | 95 | <0.01 | OR‡ |
| | 27 | Malignant diagnosis | 0.0556 | 0.0377 | 0.0735 | 81 | <0.01 | RD‡ |

*Surveyed centers/operators.

†Estimates are during pandemic.

‡Estimates are pandemic vs. pre-pandemic.

LCI = lower confidence interval. IRR = incidence rate ratio. MD = mean difference. OR = odds ratio. PRED = percent reduction. PR = prevalence ratio. Prev = prevalence: Prop, proportion. RD = risk difference. RR = rate ratio. PPE = personal protective equipment. NP = not provided. UCI = upper confidence interval. SMD = standardized mean difference. WMD = weighted mean difference.

cancer treatment. Most reviews mentioned that cancellations of treatment were observed, although to what extend this happened was not consistently provided (*Jammu et al., 2021*; *Pacheco et al., 2021*; *Zapała et al., 2022*; *Di Cosimo et al., 2022*; *Ferrara et al., 2022*; *Mazidimoradi et al., 2021*; *Riera et al., 2021*). However, reviews reported that these reductions were more pronounced during a lockdown. In the meta-analysis by *Teglia et al., 2022a*, it was found an overall reduction of −18.7% (95% CI, −13.3 to −24.1) in the total number of cancer treatments administered during January–October 2020 compared to the previous periods, with surgical treatment having a larger decrease compared to medical treatment (−33.9% versus −12.6%); among cancers, the largest decrease was observed for skin cancer (−34.7% [95% CI, −22.5 to −46.8 ]) (*Teglia et al., 2022b*). This difference would depend on the period, with the review reporting a U-shape for the period January–October 2020. *Lignou et al., 2022* reported that between 18th and 31st of January 2021, pediatric and noncancer surgical activities were occurring at less than a third of the rate of the previous year, while *Di Cosimo et al., 2022* reported cancellation/delays of treatment in 58% (95%CI, 48–67%; I², 98%) of centers. *Majeed et al., 2022* showed that shortage of treatment and delays and interruptions to cancer therapies in general were more common in low- and middle-income countries.

## Delayed and/or canceled screening

The results of 11 reviews (*Teglia et al., 2022a*; *Alkatout et al., 2021*; *Fancellu et al., 2022*; *Ferrara et al., 2022*; *Hesary and Salehiniya, 2022*; *Mayo et al., 2021*; *Mazidimoradi et al., 2022*; *Ng and Hamilton, 2022*; *Sasidharanpillai and Ravishankar, 2022*; *Vigliar et al., 2020*; *Bougioukas et al., 2018*) reporting on cancer screening are summarized in *Supplementary file 1e* and *Table 4*. Of these, five included a meta-analysis. Overall, reviews showed a decline in screening rates across all cancer types, and that differences by demographic area and time periods were observed; for instance, countries that implemented lockdowns showed a higher decline in screening rates. Within colorectal and gastric cancers, most reviews reported a reduction of at least 50% in number of endoscopies and gastroscopies compared to previous years. In the meta-analysis by *Teglia et al., 2022a*, while colorectal screening on average was reduced by 44.9% (95% CI, −53.8% to −36.1%) during January–October 2020, a U-shape association was observed. Within women-specific cancers, the meta-analyses showed a decrease in breast and cervical cancers screening rates of at least 40–50% (*Teglia et al., 2022a*). A meta-analysis focused on cytopathology practice showed that on average there was a sample volume reduction of 45.3% (range, 0.1–98.0%), although the results would depend on the tissue sampled (*Vigliar et al., 2020*). Similar findings were reported by *Alkatout et al., 2021*.

# COVID-19 pandemic and different aspects of cancer care

**SUMMARY** — COVID-19 pandemic has had a substantial, but heterogenous impact on cancer care, both clinically and psychosocially

**UMBRELLA REVIEW**

- 26,6% Strong evidence
- 63,3% Mixed evidence
- 10,1% Weak evidence

**51** reviews met inclusion criteria

Diagnosis delays | Treatment delays | Screening delays | Psychosocial and other outcomes

**MAIN FINDINGS**

PREVALENCE

- Depression
- Post-traumatic stress disorder
- Fear of cancer progression

PERCENT DIFFERENCE

- Delayed overall treatment
- Breast cancer screening
- Colorectal cancer screening*
- Cervical cancer screening*

-100 -80 -60 -40 -20 0 20 40 60 80 100

Estimate (95%CI)

*January–October 2020

**Figure 2.** Visual summary. CI, confidence interval.

### Reduced cancer diagnosis

Main findings of the 11 reviews (*Nikolopoulos et al., 2022*; *Majeed et al., 2022*; *Alkatout et al., 2021*; *Fancellu et al., 2022*; *Ferrara et al., 2022*; *Hesary and Salehiniya, 2022*; *Lignou et al., 2022*; *Mazidimoradi et al., 2021*; *Ng and Hamilton, 2022*; *Vigliar et al., 2020*; *Pascual et al., 2022*) providing data on reduction in cancer diagnosis are provided in *Supplementary file 1f* and *Table 4*. Reviews were consistent in reporting decreased diagnosis of new cancer cases during the pandemic, although

the reduction depended on the geographical area, the period being investigated and type of cancer. For example, there was a 73.4% decrease in cervical cancer diagnoses in Portugal during 2020, and in Italy, while there was up to 62% reduced diagnosis of colorectal cancer in 2020 compared to pre-pandemic years, the reduction was more pronounced in Northern Italy where strict lockdowns were implemented. Indeed, reviews showed that countries that implemented lockdowns measures showed the highest reduction in number of new cancer cases being diagnosed. Breast cancer diagnosis rates dropped by an estimate between 18% and 29% between 2019 and 2021 (*Ng and Hamilton, 2022*).

## Reduced uptake of HPV vaccination
There was only one review to summarize data on HPV vaccination, showing up to 96% reduction in number of vaccine doses administered in March–May 2020 among adolescents and young girls aged 9–26 years; the 1-year period reduction reported was much smaller (13%) (*Ferrara et al., 2022*).

## Psychological needs/distress
Thirteen reviews covered topics related to psychological needs and distress that patients with cancer experienced during the pandemic (*Dhada et al., 2021*; *Nikolopoulos et al., 2022*; *Zhang et al., 2022*; *Kirby et al., 2022*; *Legge et al., 2022*; *Ayubi et al., 2021*; *Jammu et al., 2021*; *Momenimo-vahed et al., 2021*; *Muls et al., 2022*; *Rohilla et al., 2021*; *Zapała et al., 2022*; *Hesary and Sale-hiniya, 2022*; *Piras et al., 2022*); the findings are summarized in *Supplementary file 1f* and *Table 4*. Reviews reported that the pandemic negatively impacted the psychosocial and physical well-being of cancer survivors and patients with cancer experienced different levels of anxiety, depression, and insomnia. In a meta-analysis, *Ayubi et al., 2021* reported an overall prevalence of depression and anxiety of 37% (95% CI, 27–47, $I^2$, 99.05) and 38% (95% CI, 31–46%, $I^2$, 99.08) in patients with cancer, respectively (*Ayubi et al., 2021*). Similar findings were reported by *Zhang et al., 2022*. Compared to controls, patients with cancer had higher anxiety level [standard mean difference (SMD 0.25 (95% CI, 0.08, 0.42)) *Ayubi et al., 2021*].

## Telemedicine
Telehealth was investigated and reported in 12 of the included reviews (*Dhada et al., 2021*; *Hojaij et al., 2020*; *Murphy et al., 2022*; *Alom et al., 2021*; *Lu et al., 2021*; *Mostafaei et al., 2022*; *Salehi et al., 2022*; *Zapała et al., 2022*; *Di Cosimo et al., 2022*; *Lignou et al., 2022*; *Pascual et al., 2022*; *Bezerra et al., 2022*); a summary of main findings is provided in *Supplementary file 1h*. *Salehi et al., 2022* reported that telemedicine use in breast cancer patients was the most common investigated in studies exploring cancer-specific use of telemedicine. Telemedicine was used for various reasons, with provision of virtual visit services and consultation being the most common (*Salehi et al., 2022*). One study explored various symptom tracking apps for patients with cancer, available in the mobile health market, and found that only a limited number of apps exist for cancer-specific symptom tracking (27%) (*Lu et al., 2021*). In addition, of the 41 apps found, only one was tested in a clinical trial for usability among patients with cancer (*Lu et al., 2021*). While little research exists on how patients perceived telemedicine during the COVID-19 pandemic, early data showed that majority of patients found tele-medicine service helpful and that obtaining a telemedicine service helped solve their health problem. Nevertheless, there were concerns that use of telehealth for people with cancer suggests a greater proportion of missed diagnoses (*Lignou et al., 2022*), and that telemedicine cannot be a substitute for face-to-face appointments (*Mostafaei et al., 2022*).

## Financial distress and social isolation
Five reviews reported the economic impact of COVID-19 and social isolation of patients with cancer during the pandemic (*Supplementary file 1i*; *Dhada et al., 2021*; *Kirby et al., 2022*; *Legge et al., 2022*; *Jammu et al., 2021*; *Piras et al., 2022*). While there is little research on this topic, overall, the reviews suggested financial distress with direct and indirect costs burden and social isolation being a common issue for patients with cancer. Reviews also were consistent in reporting social isolation and loneliness among patients with cancer. Several factors contributed to social isolation, including fear of infection, social distancing measures, not having visitors and lack of social interaction during treatment.

## Tobacco use and cessation

There was only one systematic review and meta-analysis to explore tobacco use and cessation during the pandemic (*Sarich et al., 2022*). Based on data from 31 studies, *Sarich et al., 2022* found that, compared to pre-pandemic period, the proportion of people smoking during the pandemic was lower (pooled prevalence ratio of 0·87 (95%CI, 0·79–0·97)). In addition, there was similar proportions among smokers before pandemic who smoked more or smoked less during the pandemic, and on average 4% (95% CI, 1–9%) reported stopping smoking. 2% reported starting smoking during the pandemic. High heterogeneity was observed across the meta-analyses results.

## Other aspects of cancer care

Eighteen reviews (*Donkor et al., 2021*; *Gascon et al., 2022*; *Hojaij et al., 2020*; *Gadsden et al., 2022*; *Majeed et al., 2022*; *Adham et al., 2022*; *Alom et al., 2021*; *Moujaess et al., 2020*; *Pacheco et al., 2021*; *Rohilla et al., 2021*; *Di Cosimo et al., 2022*; *Lignou et al., 2022*; *Pararas et al., 2022*; *Tang et al., 2022*; *Thomson et al., 2020*; *de Bock et al., 2022*; *Pascual et al., 2022*) reported on mitigations strategies and cancer service restructuring, impact of measures on cancer prognosis, and on quality of recommendations provided during COVID-19 for cancer care; findings are summarized in *Supplementary file 1j*. In the meta-analysis by *Di Cosimo et al., 2022* routine use of PPE by patient and healthcare personnel was reported by 81% and 80% of centers, respectively; systematic SARS-CoV-2 screening by nasopharyngeal swabs was reported by only 41% of centers (*Di Cosimo et al., 2022*). Five reviews also reported on potential impact of mitigation strategies on cancer outcomes/prognosis (*Alkatout et al., 2021*; *Lignou et al., 2022*; *Pararas et al., 2022*; *Tang et al., 2022*; *de Bock et al., 2022*). It was estimated that 59,204–63,229 years of life lost might be attributable to delays in cancer diagnosis alone because of the first COVID-19 lockdown in the UK, albeit the findings were based on single study. Delayed cancer screening was estimated to cause globally the following additional numbers of cancer deaths secondary to breast, esophageal, lung, and colorectal cancer, respectively: 54,112–65,756, 31,556–32,644, 86,214–95,195, and 143,081–155,238 (*Alkatout et al., 2021*). *Tang et al., 2022*, *de Bock et al., 2022* found no deterioration in the surgical outcomes of all types of cancer or colorectal cancer surgery: also no reduction in the quality of cancer removal was observed. Similar findings were also reported by *Pararas et al., 2022*, despite the number of patients presenting with metastases during the pandemic was significantly increased. *Thomson et al., 2020*, by exploring recommendations for hypofractionated radiation therapy, found that in general the recommendations during the pandemic were based on lower quality of evidence than the highest quality routinely used dose fractionation schedules.

## Discussion

The current umbrella review summarized and appraised systematically the evidence on the extent to which several aspects of cancer care were disrupted during the COVID-19 pandemic. The summary message provided by 51 systematic reviews is that there have been modifications, delays and cancellation of treatment, delays and cancellation in cancer screening and diagnosis, and patients with cancer may have experienced additional psychological, social, and financial distress. Nevertheless, appraisal of the impact of COVID-19 on cancer care is mainly based on limited and low-quality evidence, and that data mainly derive from high-income countries, with little understanding of consequences of COVID-19 on cancer care in low- and middle-income countries. In addition, limited evidence exists on whether disruptions in cancer care during the pandemic had adverse impact in prognosis of patients with cancer and mortality.

Several guidelines were provided for cancer care during the pandemic, including recommendations on mitigation strategies to prevent SARS-CoV-2 infection and cancer treatment modalities. Nevertheless, most recommendations were based on expert opinions, and little quantitative evidence was provided to support them. This aspect was highlighted also in the systematic review by *Thomson et al., 2020*. The authors explored recommendations for hypofranctionated radiation therapy before and during pandemic and found that during the pandemic there was a significant shift from established higher-quality evidence to lower-quality evidence and expert opinions for the recommended hypofractionated radiation schedules. Similar findings were reported also by *Garg et al., 2020*,

suggesting not only guidelines were based on low level of evidence, but also there was significant discordance for the role and timing of surgery, especially in early tumors.

Specific recommendations established from the guidelines such as prioritization of high-grade malignancy, as well as other aspects such as lockdowns, social restrictions, restructure of cancer care with prioritization of high-risk malignancies and use of telemedicine, fear of infection, financial distress and shortage in medications could explain the delays and cancellation in cancer treatment, screening, and diagnosis reported in several studies. For example, *Lignou et al., 2022* raised concerns that use of telehealth for people with cancer suggests a greater proportion of missed diagnoses. Most of examined systematic reviews reported a substantial reduction in treatment, screening, and diagnosis of several cancers during the pandemic, which was more pronounced for countries that implemented a lockdown. In addition, differences were observed by geographical area, suggesting that the impact on cancer treatment, screening and diagnosis could depend on mitigation strategies countries implemented as well as on country-specific healthcare organization and resources. For example, shortage of treatment and delays and interruptions to cancer therapies in general were more pronounced in low- and middle-income countries (*Majeed et al., 2022*). The findings on disruption of cancer treatment, screening, and diagnosis are in line with findings reported for other chronic diseases, such as cardiovascular disease (*Williams et al., 2021*), suggesting the adverse impact might not be cancer specific. Future research should explore and compare how different chronic diseases were impacted.

Evidence is limited on evaluating how disruption of cancer care during COVID-19 affected prognosis of patients with cancer. Limited evidence showed that the number of patients presenting with metastases during the pandemic was significantly increased, and emergency presentations and palliative surgeries were more frequent during the pandemic (*Pararas et al., 2022*). No deterioration in the surgical outcomes of colorectal cancer surgery including mortality or reduction in the quality of cancer removal was observed (*Pararas et al., 2022*; *Tang et al., 2022*). A study (*Maringe et al., 2020*) in UK estimated that 59,204–63,229 years of life lost might be attributable to delays in cancer diagnosis alone because of the first COVID-19 lockdown, but estimates were based on modeling. Several studies *Cui et al., 2022*; *Smith et al., 2021* have shown a decline in elective cancer such as colorectal cancer, despite findings showing that gastrointestinal cancer surgery during pandemic is safe with appropriate isolation measures and no delays should be implemented for both early and advanced cancer (*Sozutek et al., 2021*). A recent meta-analysis (*Whittaker et al., 2021*) showed that delaying colorectal cancer longer than 4 weeks could be associated with poorer outcomes.

Several studies and systematic reviews thereof have investigated the impact of the pandemic on psychological well-being, financial distress, and social isolation of patients with cancer, as well as the role of telemedicine in cancer care. While studies suggested depression, anxiety, post-traumatic disorder, insomnia, and fear of cancer progression being highly reported by cancer patients with estimates reaching beyond 50%, high heterogeneity was observed, and in general systemic analysis comparing the findings with pre-pandemic period rates was lacking. The pandemic was reported to have financial burden on cancer patients with direct and indirect costs. Social isolation was commonly reported and mainly driven by fear of infection, social distancing measures, and lack of social interaction during treatment. Nevertheless, there was limited effort to quantify social isolation and economic impact on cancer care. Telemedicine and remote consultations were sharply increased in use for different aspects of cancer care, including treatment, screening, and rehabilitation. However, evidence is limited in evaluating and quantifying the positive and negative impact, as well as cost-effectiveness of telemedicine. While limited evidence suggested telemedicine reduced costs of cancer care for both patients and healthcare provider, there were concerns especially from patients that telemedicine could not have similar benefits to on-site consultations.

Our study has certain limitations. Although our search was based on recent recommendations on optimal databases needed to be searched for umbrella reviews (*Goossen et al., 2020*), we cannot rule out missing some other relevant systematic reviews. Most systematic reviews included in this umbrella review were based on intermediate and high risk of bias studies, and the findings were mainly based on case-series, cross-sectional and retrospective observational study designs which are prone to residual confounding and poor in determining temporal associations. Prevalence and incidence estimates are also subject to selection biases. In some instances, data were derived from one study or from studies with small sample sizes and limited number of events, leading to large uncertainty. Many studies did not include any pre-pandemic controls. Furthermore, some of the evidence overlapped

among the systematic reviews that were included in this umbrella review, but this allows comparing notes on results and conclusions for the overlapping efforts. Some systematic reviews were published early (in 2020), and thus they had even more limited evidence and the impact of the disruptions may have differed across different pandemic waves. Most findings were derived from high-income and/ or western countries, limiting the generalizability of the findings to low- and middle-income countries. Lastly, concreate conclusions on intermediate, and long-term impact remain unclear. Finally, the suboptimal methodological rigor of many included reviews is notable.

In summary, evidence shows a diverse and substantial impact of the COVID-19 pandemic on cancer care, including delays in treatment, screening, and diagnosis. Also, patients with cancer had been affected psychologically, socially, and financially during the COVID-19 crisis. However, large uncertainty and gaps exist in the literature on this topic. Most of the evidence on the topic is derived mainly from high- and middle-income countries, and low-quality studies, and thus, future high-quality studies with larger geographical capture and properly performed, rigorous systematic reviews with careful meta-analyses will continue to have value in this field.

## Materials and methods

We performed an umbrella review following the recent published guideline (*Belbasis et al., 2022*), and for reporting we adhered to the Preferred Reporting Items for Overviews of Reviews – PRIOR checklist (*Gates et al., 2022*; *Supplementary file 1k*). The protocol has been registered with the Open Science Framework (https://osf.io/qjgxv).

### Search strategy

Literature search was performed in PubMed and WHO COVID-19 database using the search strategy in *Supplementary file 1l*. No language restriction was applied. We searched for studies published until November 3, 2022; an update of the search was performed until November 29, 2022. References cited in the final included studies for analysis were further screened to identify other relevant publications.

### Screening, study selection, and eligibility criteria

Retrieved items were first screened based on the title and abstract and potentially eligible references were then screened in full text. Screening was performed by two reviewers and in case of discrepancies, a final decision to include or exclude was settled with discussion. We included studies if they fulfilled all the following criteria: (1) were systematic reviews with our without meta-analysis or individual participant meta-analysis; (2) included individuals diagnosed with any type of cancer and at any cancer stages (early to advanced), or individuals targeted for cancer screening; (3) assessed the impact of the COVID-19 pandemic, and thus had data collected during the pandemic period (2020–2022) (the included studies may nevertheless have used also control pre-pandemic periods in order to assess the magnitude of change during the pandemic); and assessed any of the following outcomes: delay/cancellation of treatment (overall and per specific treatment); modification of treatment (overall and per specific treatment); delayed/canceled screening (overall and per specific type of screening); reduced diagnoses (overall and per specific diagnosis); psychological needs; ethical needs; social needs; financial burden and distress; social impact/isolation; psychological distress; use of telehealth/virtual visits, and other aspects of cancer care such as impact of the COVID-19 pandemic on prognosis. In addition, irrespective of including patients with cancer, we included reviews that looked at impact of COVID-19 on uptake of HPV vaccination and tobacco use and cessation.

### Data extraction and critical appraisal

The data extraction was performed by one of the authors and the extracted data were further checked by two other authors; differences were settled by discussion. In case an eligible article included data from several diseases, when feasible, we extracted information only on cancer-related outcomes of our interest. First, we extracted general information from the eligible reviews, including information on authors, year of publication, type of studies considered (design), number of eligible studies, COVID-19 period covered (until when), whether it has considered studies with pre-pandemic controls (yes exclusively/yes for some/not at all), the outcomes examined and for which cancers

each outcome was examined, and methods of analysis and heterogeneity (if provided). To provide the geographical breadth of the evidence, we extracted information on location(s) of the individual studies included in the eligible reviews; for example, retrieving information on countries and areas or whether the studies were done in multiple countries. Concerning the methodological rigor, for each systematic review we extracted information on whether the authors used any previously validated tool or any other set of extracted items to assess the methodological rigor of the included studies. If yes, we recorded the tool used and the main conclusions of the assessment were grouped in the broad categories: most studies were weak in methodological rigor, most studies were strong in methodological rigor, or mixed/intermediate pattern between the other two categories. Two reviewers assessed methodological rigor of the included systematic reviews using the AMSTAR-2 tool (*Shea et al., 2017*); any discrepancies were settled with the help of a third reviewer. AMSTAR-2 is based on a 16 item or domain checklist, with seven of these items considered critical for the overall validity of a review. The domains considered critical are: (1) protocol registration before starting the review; (2) adequate and comprehensive search of the literature; (3) providing justification for the exclusion of individual studies; (4) risk of bias assessment of the studies included in the review; (5) use of appropriate statistical methods in performing a meta-analysis; (6) accounting for risk of bias when interpreting the results; (7) evaluation of the presence and impact of publication bias. Last, based on abstract and full-text reading, we extracted information on main conclusions derived from each of the included reviews. When the review included several disease areas, we extracted information on main findings of the included individual studies within the review that were relevant to cancer.

## Statistical analysis

Due to high heterogeneity in the designs, study questions, outcomes, and metrics, a descriptive analysis was performed. We calculated the proportion of reviews that provided information on single countries and multiple countries. Median and interquartile range were calculated for some of the characteristics of the eligible reviews (e.g., number of databases searched). Separate tables were created for the methodological appraisal of the systematic reviews, the methodological appraisal of the studies in each systematic review, for the characteristics and subject matter information of each systematic review, and for the final conclusions of each systematic review. In addition, we created a separate table for reviews that implemented meta-analysis, providing the summary estimates, 95% CIs, and heterogeneity estimates. Limitations and areas of limited evidence were noted.

## Acknowledgements

We would like to thank Beatrice Minder for helping with search strategy and Dr. Erand Llanaj (Department of Molecular Epidemiology, German Institute of Human Nutrition Potsdam-Rehbruecke, Nuthetal, Germany) for designing and illustrating the graphical abstract.

## Additional information

### Funding

No external funding was received for this work.

### Author contributions

Taulant Muka, Conceptualization, Data curation, Formal analysis, Validation, Investigation, Methodology, Writing – original draft, Writing – review and editing; Joshua JX Li, Sahar J Farahani, Conceptualization, Data curation, Formal analysis, Writing – review and editing; John PA Ioannidis, Conceptualization, Data curation, Formal analysis, Supervision, Validation, Investigation, Methodology, Writing – original draft, Writing – review and editing

### Author ORCIDs

Taulant Muka http://orcid.org/0000-0003-3235-3073
John PA Ioannidis http://orcid.org/0000-0003-3118-6859

Decision letter and Author response
Decision letter https://doi.org/10.7554/eLife.85679.sa1
Author response https://doi.org/10.7554/eLife.85679.sa2

## Additional files

### Supplementary files
- Supplementary file 1. (1a-l) Table characteristics, main findings, PRISMA, and search strategy.
- Supplementary file 2. Bibliographic databases used from each review (see excel file).
- MDAR checklist

### Data availability
All data are in the manuscript and supplements.

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
