## [Editor Report]

This solid work reviews and synthesizes evidence of the impact of the COVID-19 pandemic on a variety of cancer outcomes. The results have potentially important implications for various fields of cancer research as they review evidence spanning from cancer prevention efforts to changes in diagnoses and cancer treatment modalities.

---

## [Decision Letter]

**Decision letter after peer review:**

Thank you for submitting your article "Changes in cancer prevention and management and patient needs during the COVID-19 pandemic: An umbrella review of systematic reviews" for consideration by *eLife*. Your article has been reviewed by 2 peer reviewers, one of whom is a member of our Board of Reviewing Editors, and the evaluation has been overseen by a Reviewing Editor and Eduardo Franco as the Senior Editor. The following individual involved in the review of your submission has agreed to reveal their identity: Neda Kabiri (Reviewer #2).

Essential revisions:

1. Clarify the objectives of the review.

2. Revise Table 3 to clarify column headings.

3. Address the searching of only one database as a major limitation of review.

*Reviewer #1 (Recommendations for the authors):*

The review appears to me to be somewhat unfocused, and it is not clear to me how analyses and extracted data align with the authors' objectives. I think this could be improved by the following:

– In the introduction, the authors could more clearly state why an umbrella review is needed now, and what an umbrella review can teach us that individual systematic reviews cannot, i.e. what they hope to learn.

– The objective of "synthesis" is very vague. I would suggest listing more specific objectives for the umbrella review so the reader can better assess the authors' interests (ex. assess the quality of systematic reviews of COVID-19 impact, assess the geographical breadth of the available evidence, summarize conclusions from different reviews, assess evidence gaps, etc.)

– Explain how the extracted data relates to the above objective(s), and how the analyses performed relate to the above objective(s).

– Structure the results and Discussion section in line with the above objective(s).

The inclusion of various health outcomes of interest is laudable but makes it more challenging to present a coherent picture across different outcomes as very little space is dedicated to each. This is a suggestion, but I think this could potentially be improved by separating out the studies into different tables stratified by the outcome and presented in their relevant section in the results, rather than combining all studies into larger tables mixing very different outcomes.

Tables are over-reliant on acronyms. While I understand space is always an issue in tables, they would be much more readable if fewer acronyms were used. If keeping acronyms, please check legends as acronyms were not always defined.

Table 3 is not self-sufficient as readers do not know the topic of each question in AMSTAR-2, so the table is uninformative without a copy of the tool. The authors may want to consider defining questions in the footnotes or providing short titles to questions rather than numbers.

Systematic reviews of HPV vaccination and tobacco cessation do not appear to fit the eligibility criteria, where the target population is defined as "included individuals diagnosed with any type of cancer and at any cancer stages (early to advanced), or individuals targeted for cancer screening". Please revise.

The protocol registry the authors have used appears to be more adapted to observational studies than reviews. In the future, the authors may want to consider using registries that are more suited to review protocols (ex. PROSPERO).

*Reviewer #2 (Recommendations for the authors):*

I appreciate the effort by the authors to conduct an umbrella review for assessing the Changes in cancer prevention and management and patient needs during the COVID-19 pandemic. While the findings are very important, I believe they will be best served with major revisions to the manuscript prior to publication.

The title does not reflect what you have done in your review. In my idea 'the impact of COVID-19 on cancer care' which you have mentioned in the objective is better.

Line 26: how you quantified the impact of …? As you had a descriptive analysis.

As a habit, we need to avoid the language construct of "cancer patients" (many places in the manuscript); patients don't want to be defined as people by their illness. Better to say "patients with cancer". Line 35: forty five.

Line 288: only appraises?

What is your final conclusion? Explain it more in detail.

Figure 1. You have mentioned 'no English language' as a reason for the exclusion of one study. Since you had "No language restriction was applied." In the methods.

Appendix 2 search strategy: did not you use any MeSH terms? How about synonyms for 'cancer'? as you have included a meta-synthesis in your review, was not it better to include this word in the search strategy? And did you use (systematic review OR meta-analysis) as a keyword to search or as a filter in PubMed search? Although you have conducted your search, was it enough to search only one database? What was the logic behind it? Which you have mentioned this as a major limitation of your study.

There is no need for appendix 5 since the data is repeatedly presented in appendix 3.

You can merge appendices 6-13 in one.

A careful review of English language usage and grammar is required before publication.

---

## [Author Response]

Reviewer #1 (Recommendations for the authors):The review appears to me to be somewhat unfocused, and it is not clear to me how analyses and extracted data align with the authors' objectives. I think this could be improved by the following:– In the introduction, the authors could more clearly state why an umbrella review is needed now, and what an umbrella review can teach us that individual systematic reviews cannot, i.e. what they hope to learn.

We agree with the reviewer that this is an important aspect to clarify. We have now revised the introduction to address the need of doing an umbrella review.

“While several systematic reviews have examined the impact of COVID-19 on cancer care, they evaluated different outcomes and periods of the pandemic, and thus the available review findings are rather fragmented 3 4 8-14. A comprehensive review of impact of COVID-19 on several aspects of cancer would be essential to understand gaps and scale-up evidence-based interventions, including learning lessons for future pandemics. In addition, although systematic reviews are important for public health and policy decision-making during the pandemic, the level of methodological rigor they implemented is unclear.”

– The objective of "synthesis" is very vague. I would suggest listing more specific objectives for the umbrella review so the reader can better assess the authors' interests (ex. assess the quality of systematic reviews of COVID-19 impact, assess the geographical breadth of the available evidence, summarize conclusions from different reviews, assess evidence gaps, etc.)

We share the reviewer’s concerns. We have now revised the aim to clarify the objectives of the umbrella review.

“In the current study we performed an umbrella review of systematic reviews to summarize the impact of COVID-19 on several aspects of cancer care, including treatment, diagnosis, financial, psychological and social dimensions. We assessed the amount and geographical breadth of the available evidence and methodological rigor of the primary studies included in each review (as assessed by the reviewers) and of the systematic reviews themselves; and summarized the conclusions from different reviews on COVID-19 impact.”

– Explain how the extracted data relates to the above objective(s), and how the analyses performed relate to the above objective(s).

We followed the reviewer’s suggestion and now organized the data extraction section according to our objectives, and provided information on how such information was extracted and analyzed in the analysis section.

“The data extraction was performed by one of the authors and the extracted data were further checked by two other authors; differences were settled by discussion. In case an eligible article included data from several diseases, when feasible, we extracted information only on cancer-related outcomes of our interest. First, we extracted general information from the eligible reviews, including information on authors, year of publication, type of studies considered (design), number of eligible studies, COVID-19 period covered (until when), whether it has considered studies with pre-pandemic controls (yes exclusively/yes for some/not at all), the outcomes examined and for which cancers each outcome was examined, and methods of analysis and heterogeneity (if provided). To provide the geographical breadth of the evidence, we extracted information on location(s) of the individual studies included in the eligible reviews; for example, retrieving information on countries and areas or whether the studies were done in multiple countries. Concerning the methodological rigor, for each systematic review we extracted information on whether the authors used any previously validated tool or any other set of extracted items to assess the methodological rigor of the included studies. If yes, we recorded the tool used and the main conclusions of the assessment were grouped in the broad categories: most studies were weak in methodological rigor, most studies were strong in methodological rigor, or mixed/ intermediate pattern between the other two categories. Two reviewers assessed methodological rigor of the included systematic reviews using the AMSTAR-2 tool17; any discrepancies were settled with the help of a third reviewer. AMSTAR-2 is based on a 16 item or domain checklist, with seven of these items considered critical for the overall validity of a review. The domains considered critical are: (i) protocol registration before starting the review; (ii) adequate and comprehensive search of the literature; (iii) providing justification for the exclusion of individual studies; (iv) risk of bias assessment of the studies included in the review; (v) use of appropriate statistical methods in performing a meta-analysis; (vi) accounting for risk of bias when interpreting the results; (vii) and evaluation of the presence and impact of publication bias. Last, based on abstract and full text reading, we extracted information on main conclusions derived from each of the included reviews. When the review included several disease areas, we extracted information on main findings of the included individual studies within the review that were relevant to cancer.”

“Due to high heterogeneity in the designs, study questions, outcomes, and metrics, a descriptive analysis was performed. We calculated the proportion of reviews that provided information on single countries and multiple countries. Median and interquartile range were calculated for some of the characteristics of the eligible reviews (e.g., number of databases searched). Separate tables were created for the methodological appraisal of the systematic reviews, the methodological appraisal of the studies in each systematic review, for the characteristics and subject matter information of each systematic review, and for the final conclusions of each systematic review. In addition, we created a separate table for reviews that implemented meta-analysis, providing the summary estimates, 95% confidence intervals and heterogeneity estimates. Limitations and areas of limited evidence were noted.”

– Structure the results and Discussion section in line with the above objective(s).

The result section reflects now with the revised objectives. In the discussion we have put more emphasizes on main findings related to COVID-19 impact on cancer care, as well as recognized the limitations that come from little representations of data from low and middle income countries, as well as from the low quality of evidence.

The inclusion of various health outcomes of interest is laudable but makes it more challenging to present a coherent picture across different outcomes as very little space is dedicated to each. This is a suggestion, but I think this could potentially be improved by separating out the studies into different tables stratified by the outcome and presented in their relevant section in the results, rather than combining all studies into larger tables mixing very different outcomes.

We agree with the reviewer that data presentation is challenging when dealing with different outcomes. However, for the main text, we compressed the information to provide a general information, but in the supplemental material we provide separate tables for each outcome.

Tables are over-reliant on acronyms. While I understand space is always an issue in tables, they would be much more readable if fewer acronyms were used. If keeping acronyms, please check legends as acronyms were not always defined.

Due to various outcomes and data we included, we tried to condense the information in main tables to provide as much as information as possible. We now revised the legends, so all acronyms are defined and are easy to read.

Table 3 is not self-sufficient as readers do not know the topic of each question in AMSTAR-2, so the table is uninformative without a copy of the tool. The authors may want to consider defining questions in the footnotes or providing short titles to questions rather than numbers.

Thank you for the excellent feedback. We have now added the questions in the Footnote.

Systematic reviews of HPV vaccination and tobacco cessation do not appear to fit the eligibility criteria, where the target population is defined as "included individuals diagnosed with any type of cancer and at any cancer stages (early to advanced), or individuals targeted for cancer screening". Please revise.

We share the reviewer`s concern. We have now clarified this aspect of inclusion criteria.

“In addition, irrespective of including patients with cancer, we included reviews that looked at impact of COVID-19 on uptake of HPV vaccination and tobacco use and cessation.”

The protocol registry the authors have used appears to be more adapted to observational studies than reviews. In the future, the authors may want to consider using registries that are more suited to review protocols (ex. PROSPERO).

We thank the reviewer for the suggestion. We will make sure next time we submit in a register that is more suitable to review protocols.

Reviewer #2 (Recommendations for the authors):I appreciate the effort by the authors to conduct an umbrella review for assessing the Changes in cancer prevention and management and patient needs during the COVID-19 pandemic. While the findings are very important, I believe they will be best served with major revisions to the manuscript prior to publication.The title does not reflect what you have done in your review. In my idea 'the impact of COVID-19 on cancer care' which you have mentioned in the objective is better.

We followed the reviewer’s suggestion and revised the title as following “The impact of the COVID-19 pandemic on cancer prevention and management, and patient needs: An umbrella review of systematic reviews”.

Line 26: how you quantified the impact of …? As you had a descriptive analysis.

This is an important point to clarify. As highlighted in the paper, we have extracted also information on reviews that performed meta-analysis and compared data with pre-pandemic period. We have now added this aspect in the statistical analysis.

“In addition, we created a separate table for reviews that implemented meta-analysis, providing the summary estimates, 95% confidence intervals and heterogeneity estimates. Limitations and areas of limited evidence were noted.”

As a habit, we need to avoid the language construct of "cancer patients" (many places in the manuscript); patients don't want to be defined as people by their illness. Better to say "patients with cancer". Line 35: forty five.

We agree with the reviewer. We have now revised throughout the manuscript.

Line 288: only appraises?

We revised the sentence to also add “summarize”.

“The current umbrella review summarized and appraised systematically the evidence on the extent to which several aspects of cancer care were disrupted during the COVID-19 pandemic”

What is your final conclusion? Explain it more in detail.

We revised the conclusion as following:

“In summary, evidence shows a diverse and substantial impact of the COVID-19 pandemic on cancer care, including delays in treatment, screening and diagnosis. Also, patients with cancer had been affected psychologically, socially, and financially during the COVID-19 crisis. However, large uncertainty and gaps exist in the literature on this topic. Most of the evidence on the topic is derived mainly from high and middle-income countries, and low-quality studies, and thus, future high-quality studies with larger geographical capture and properly performed, rigorous systematic reviews with careful meta-analyses will continue to have value in this field.”

Figure 1. You have mentioned 'no English language' as a reason for the exclusion of one study. Since you had "No language restriction was applied." In the methods.

We clarified this issue in the footnote of the figure.

“In the search, we did not include any language restriction filter. However, during full text screening we included only studies that were in English.”

Appendix 2 search strategy: did not you use any MeSH terms? How about synonyms for 'cancer'? as you have included a meta-synthesis in your review, was not it better to include this word in the search strategy? And did you use (systematic review OR meta-analysis) as a keyword to search or as a filter in PubMed search? Although you have conducted your search, was it enough to search only one database? What was the logic behind it? Which you have mentioned this as a major limitation of your study.

We agree with the reviewer’s remarks and suggestions. As clarified in our response to the first comment, we addressed this issue by having the search strategy designed for a specialized librarian. We now consider our approach as a strength of our study.

There is no need for appendix 5 since the data is repeatedly presented in appendix 3.

We share the reviewer`s concern. However, we added Appendiy 5 to have a better representation of data (derived from Appendix 3) on methodological rigor. Therefore, we decided to keep the table.

You can merge appendices 6-13 in one.

Mering appendices 6-13 could be a good idea. However, since there were different outcomes, it could be easier to read and follow having separate tables for each outcome.

A careful review of English language usage and grammar is required before publication.

We have now revised the manuscript throughout for English language.